

**Rising bubbles as mechanism for scavenging and aerosolization of diatoms**
**Roman Marks[1], Ewa Górecka[2] Kevin Mc Cartney[3], Wojciech Borkowski[1]**
[1] University of Szczecin, Faculty of Geosciences, Physical Oceanography Unit,
6        Mickiewicza 16, 70-383 Szczecin, Poland;
[2] Natural Sciences Education and Research Centre, University of Szczecin,
8        Palaeoceanology Unit, Mickiewicza 16a, 70-383 Szczecin, Poland;
[3] Department of Environmental Sciences and Sustainability, University of Maine at Presque
10       Isle, Presque Isle, ME  04769, USA.

*Correspondence to:* Roman Marks (Roman.Marks@usz.edu.pl)
**Abstract.** Bubbles rising in clean saline water cause steady displacement of ions at the bubble
boundaries that separate anions and cations based on ion mass. Anions of greater mass are
resistant to displacement and concentrate on the bubble upper half sphere, while smaller and
less massive cations are displaced towards lower pressure of the bottom half sphere. The
separation into anionic and cationic domains on the bubble curvatures creates electric polarity
that may draw particulates dispersed in the water. Viable diatoms as well as bacteria develop
negative charge on outer membranes, that are attracted to the cationic bubble bottom half
sphere and pocket. When bubble bursts at the air/water interface the diatoms and bacteria are
ejected into the air with initial or secondary jet droplets that are projected upward with a small
water column derived from a cationic vortex. Experiments conducted in brackish and oceanic
saline water on *Nanofrustulum* and *Cyclotella cells* indicated that the averaged concentration
in jet droplets compared to the original water volume (here termed the enrichment factor) for
aerosolized diatoms may range from 8 to 307.

**1    Introduction**
**1.1    Rising bubbles**
Bubbles in the ocean are abundantly caused by breaking waves in surface waters (Woodcock,
1953; Blanchard and Syzdek, 1970; Monahan et al., 1983; Garbalewski and Marks, 1987;
Czerski et al., 2011) and rain drop impact upon the water surface (Blanchard and Woodcock,
1957; Marks, 1990). Bubble plumes are diffused by near surface turbulence to depths of four
to six times the significant wave height (Thorpe, 2001). Production of bubbles increases with
wind speed (Blanchard, 1963; Monahan et al., 1983; Callaghan et al., 2007) and effervescence
bubbles increase with dissolved gas supersaturation (Stramska et al., 1990; Marks, 2008)
which is common in shallow water environments, especially, when phytoplankton production





of oxygen occurs during spring warming and upwelling (Blanchard, 1963; Garbalewski and
Marks, 1987). Under such conditions enhanced formation of small bubbles and sea-derived
aerosols may occur (Garbalewski and Marks, 1987; Stramska et al., 1990; Marks, 2008).

Bubbles produced by breaking waves in oceanic water range from radius 1 to several

thousand μm with maximum occurrence about 25 μm (Woolf, 1997). Although wave break
agitation is the main mechanism of bubble generation in fresh and oceanic waters, only saline
waters generate abundant small bubbles (Thorpe, 2001; Woolf, 1997; Czerski et al., 2011).
Recent research indicates that small bubbles with radii less than 30 μm are especially
abundant and have significant effect on upper ocean optical properties (Stramski and
Tęgowski 2001; Czerski et al., 2011). A typical breaking wave generates a downward
circulating rotor–like motion reaching a depth equal to the wave height. Thus a surface wave
of 1 m height, generated at wind velocity 8 m/s, can produce and disperse bubbles to about 1
m water depth (Thorpe, 2001) while during rain bubbles may occur to about 10-20 cm depth
(Blanchard, 1963; Katsaros and Buettner, 1969). However, local generation of splash droplets
and bubbles is further enhanced during high and in particular tropical precipitation
(Garbalewski and Marks, 1987). Higher water temperature decreases water viscosity (Woolf,
1997) which enhances  bubbles and sea salt aerosol production.

After downward dispersion in the water column, bubbles tend to surface according to

bubble volume that controls buoyant vertical motion, and gradually grow in size, especially
under the condition of dissolved gases supersaturation (Woolf, 1997; Marks, 2008). Field and
laboratory experiments confirm that bubble production in oceanic water and generation of sea
salt aerosols increases with increasing degree of dissolved gases saturation (Stramska et al.,
1990; Marks, 2008). In addition, bubble production increases with water temperature and
water to air thermal gradient (Marks, 1987).

Experiments on the rotational features of a rising bubble (Marks, 2014) suggest that

interaction with ions in "clean or relatively clean" sea water can separate cations and anions
into charge polarized and oppositely rotating domains. The principle on which ions are
separated by rising bubbles is based on mass differences between the main ionic hydrates ($Cl^-$
and $Na^+$) that compose sea water. Using a Cl/Na atomic mass equal to 1.542 (Kropman and
Bakker, 2001) the bubble mediated selection of ions that collide with bubble outer curvatures
may be based on ionic hydrate mass differences. Anionic hydrates are more resistant to
displacement and concentrate on the upper bubble half sphere, while lighter and smaller
cationic hydrates are drawn to a low pressure area, associated with the bubble bottom half
sphere and sub-bubble vortex (Marks, 2014; 2015). Ionic motion accelerated at the moment of



interception with a bubble boundary may exceed 10 times the gravitational acceleration,
which generates opposite directed rotations, anionic-dextrorotary (a-dx) on the bubble upper
curvature and cationic-levorotary (c-lv) on the bubble bottom half sphere (Marks, 2014).

**1.2   Bubble mediated scavenging of bio-cells**
First evidences of bubble mediated bio-cells scavenge and incorporated into jet droplets was
reported by Blanchard and Syzdek (1970), who investigated the aerosolization of bacteria
*Serratia marcescence* by uniform - sized bubbles in water columns of different heights. The
bacteria were collected and incubated on agar plates and counted. The number of bacteria
cells ejected with jet droplets ($Njd$) rated to bacteria concentration in the same  (as in droplets)
volume of water in which the bubbles rise/burst ($Nw$) was used to estimate the "enrichment
factor- ($Ef)$" in a form:
$$Ef = Njd / Nw. \qquad\qquad (1)$$
Blanchard and Syzdek (1982) determined $Ef$ values up to 600, in experiments were the
bubble rise distance in the water column was 10 cm. More recent experiments reported that
($Ef$) values of bacteria in aerosol droplets range from one order of magnitude for oceanic
waters (experiments conducted by Aller et al., 2005) to three orders for brackish coastal
waters (Marks et al., 2001). Moreover, the experiments indicated that the sea - derived
droplets may also contain a significant share of fungi (Marks et al., 2001) and viruses
(Matthais-Maser and Jaenicke, 1994; Aller et al., 2005; Burrows et al., 2009). However, a
high discrepancy between level of airborne bacteria was reported by Mayol (2014) likely
associated with the method applied to bacteria counting. A modern method based on
RNA/DNA identification offers significantly higher counts that include both viable and dead
cells, while previous traditional methods counted only viable cells.
The waterborne bacteria may effectively adhere to the bubble boundary (Blanchard
and Syzdek, 1982; Weber et al., 1983). Both teams of researchers reasoned that interception
occurs when small particles move along fluid streamlines around the bubble wall, thus
particles (including bacteria) are accumulated at the bubble boundary. With increasing
distance of rise the bubble surface changes gradually from a mobile to rigid structure, which
decreases the rate of bacteria collection (Weber et al., 1983).
Until now no bubble related research has focused on diatom scavenge. Nevertheless,
airborne transport of both freshwater and marine diatom taxa was documented in various



locations including a highly elevated glaciers (Sherilyn et al., 2015) and continental interiors
such as the Antarctic continent (Budgeon et al., 2012; Stanish  et al., 2013). Thus, one
purpose for this study was to collect experimental evidences that diatom cells may be
scavenged and aerosolized.

**1.3      Bubbles burst at water-air interface**
At a clean air-water interface, bubbles of diameter ($D$) burst almost instantly upon arrival and
eject a few large jet droplets of diameter that is roughly an order of magnitude less than $D$.
The droplets are derived from the bubble lower half and hundreds of smaller film droplets
derived from upper half-bubble (Blanchard and Woodcock, 1957; Lovett, 1978; Blanchard,
1989). Sea-derived droplets supply the troposphere with various materials dominated by sea
salt aerosols that provide condensation nuclei and affect electric properties (Blanchard, 1963;
Marks, 1990). In addition, marine aerosols contain a significant share of trace elements and
hydrophobic particulates (Liss, 1983; Novakov and Penner, 1993; Duce, 2001; Tuck, 2002;
Bigg and Leck, 2008; Marks and Bełdowska 2001).

About 50% of kinetic wave energy imparted into ocean wave motions is dissipated by

bubbles (Terray et al., 1996). The bubble motions including the rotational kinetics (Marks,
2014; 2015) play a significant role in redistribution of radiant solar energy absorbed by the
near surface water. The energy dissipation continues from the moment of bubble formation in
the water column and after the bubble bursts at the sea surface, where a share of rotational
kinetic energy is transferred into jet droplets (Marks, 2014). After this process, a final
dissipation of energy to heat and evaporation occurs with duration according to evaporation
rate and air relative humidity, which over the sea is usually higher than 80%, allowing
droplets to persist in the liquid phase (Garbalewski and Marks, 1987). With respect to the
humidity, oceanic tropical belt and warm compartments near the east coast of continents are
the most efficient evaporative zones on the Earth (Yu, 2007).

Saline waters can generate bubbles and aerosols which could supply the troposphere

with abundant bio-aerosols (Burrows et al., 2009). Laboratory experiments by Blanchard and
Syzdek (1970), Weber et al. (1983) and Marks et al. (2001) indicated that jet droplets may be
highly enriched by bacteria. The ratio of bacteria in the jet droplets to that in the bulk water
varies with the rise distance in the water column and is highest during the first few up to c. 30
cm of rise (Weber et al., 1983) then decreases due to overload with respect to scavenged
cargo (Blanchard and Syzdek, 1970; Marks et al., 2001). The present research further explores



the bubble mediated scavenge and aerosolization of bio-molecules with special attention
given to diatoms.

**2      Methods**

Diatoms stains were obtained from Szczecin Diatom Culture Collection (SZCZ) curated at the
Natural Sciences Education and Research Centre, Faculty of Geosciences, University of
Szczecin. Two stains of diatoms were prepared: (1) *Nanofrustulum*   sp. (SZCZ_E_517)
forming elongated chains of about 3 by12 μm sizes in two concentration of 11000 and 480000
cells / 1 ml suspended in an artificial (f/2) medium of 35 g/kg salinity and (2) *Cyclotella* cf.
*meneghiniana* Kützing (SZCZ_E_684) with spherical shape of 10 μm diameter (Fig. 1) in
concentration of 3000 cells/1 ml, suspended in artificial (f/2) medium of 15 g/kg salinity. To
prepare the solutions an instant sea salt was used.

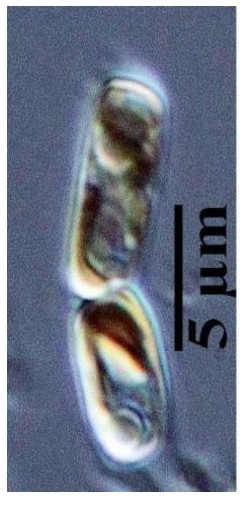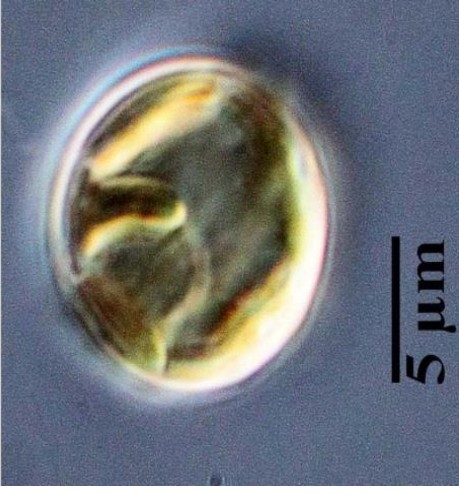

**Figure 1.** Stains of diatoms used in experiments: A) elongated *Nanofrustulum* and B)
spherical *Cyclotella*. Picture taken by Canon DS 500D using Zeiss Scope A1 with PlanApo
x100 lens.

Experiments were conducted in several stages: 1) an experimental set-up was tested
and size of bubbles produced by glass capillary aerator was measured; 2) the bubble-mediated
aerosolization of elongated *Nanofrustulum* and spherical *Cyclotella* diatom cells was
confirmed; 3) four sets of experiments using the two different diatom stains suspended in
brackish and oceanic water were conducted and evaluated; 4) twenty additional experiments
were conducted using more concentrated diatom stain suspended in water of 35 b/kg salinity.



In addition, complementary investigations were conducted on: 1) the electric charge
distribution along the bubble boundaries; 2) charge of jet droplets; 3) outer charge
incorporated to diatom membranes; 4) the ejection height of initial and secondary droplets
above water level; and 5) diatoms content in initial and secondary jet droplets using
negatively charged, vertically placed Plexiglas plate.

**2.1     Experimental procedure**

During experiments 150 ml volume of each prepared suspension was placed in a 200 ml glass
beaker and aerated using a glass capillary, producing stream of bubbles diameter $D = 1.2$ mm.
Bubbles were generated by air pump equipped with cotton filter and rose through a 10 cm
water column in a 200 ml breaker (Fig. 2). Upon arrival at the water surface a bursting bubble
produced 5 to 7 jet droplets, which were collected on a standard microscopy glass slide (26 x
76 mm) placed transversely to jet droplet motion at 4 cm above the water surface.

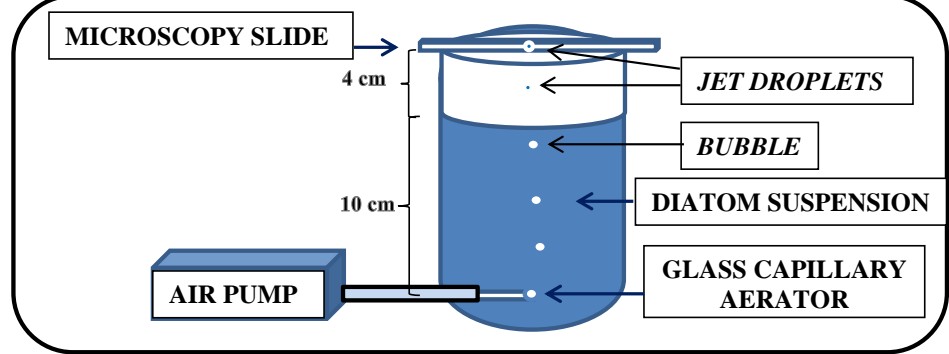

**Figure 2.** Experimental set-up used to determinate enrichment factor for aerosolized diatoms.

After 2-3 seconds of exposure microscopy slides were placed into Petri dish
containing a small volume of distilled water and covered to prevent drying. Counting of
diatom cells was conducted using Nikon Eclipse TS100 inverted light microscope with 20X
lens. Taking into account that both diatoms concentrations in the water suspension and water
salinity (or more precisely the availability of cations) decrease during aeration, only 8 samples
were collected to estimate enrichment factor for aerosolized *Cyclotella* diatoms from one
suspension of low or moderate concentrated diatom cells. In case of high diatom
concentration of *Nanofrustulum* in 35 g/kg salinity, the number of samples was extended to





20. However, after analyzing that set of data we noticed that the efficiency of diatom aerosolization increased, indicating that rising bubbles reduced concentrations of diatoms (solution was gradually cleaned) during the experiments.

In order to detect electric charge distribution around the rising bubble boundaries an oscilloscope, type HANTEK®-DS01201BV was used, allowing to trace the voltage polarity in mV with 0.1 mV detection limit and 0.5% accuracy. Similarly, the electric charge of diatoms adhered onto the oscilloscope probe was measured. The complementary observations allowed determination of ion accumulation and electric polarity on both bubble curvatures and diatom exteriors. The presence of diatoms included in both the initial and secondary jet droplets was investigated by microscopy glass plates exposed at different heights above water level. In addition, a deflection of jet droplets towards negatively charged Plexiglas plate (24 x 6 x 2 mm) was observed.

## 3 Results

### 3.1 Aerosolization of diatom stains

A single burst of bubble $D = 1.2$ mm may ejects about 4-8 droplets from a clean water at temperature (*Tw*) of 20°C (Blanchard and Syzdek, 1982); the number of jet droplets that contained diatoms were counted and compared with droplets that lacked diatom cargo. Conducted screening showed that only 20-25% of jet droplets were enriched by diatoms, which suggests that the process of diatom aerosolization might be influenced by a combination of factors operative in the water column, or at air-water interface. To explore these variables a set of experiments collected droplets at different elevations above water level to determinate whether diatoms were included in the initial large jet droplets or secondary droplets. The observations indicated that not exclusively top jet droplets were enriched by diatoms. That result suggests that perhaps a first initial droplets are projected upward too fast to skim diatoms thus the next and slower droplet that follows, hereafter called sub - initial droplet may lift diatoms.

An example of a jet droplet containing diatoms and bacteria is illustrated in Fig. 3, which shows the initial or sub-initial jet droplet of about $d = 0.12$ mm, ejected by bubble of c. $D = 1.2$ mm, at $Tw = 22.1$°C. A cargo of diatom cells in all collected initial jet droplets ranged from 1-12 for *Nanofrustulum* and 1-3 for *Cyclotella* stains indicating that elongated (*Nanofrustulum*) diatoms that offer 8% greater outer surface were subject to more enhanced scavenge as compared with spherical (*Cyclotella*) cells. In the case of *Nanofrustulum*



aerosolization, the higher salinity of 35 g/kg might contribute to enhanced scavenge and
aerosolization, since initial jet drops are ejected higher above seawater than above distilled
water (Blanchard, 1989). Since diatom cells are relatively large, as compared with bacteria
cells, these should require more energy for ejection into air.

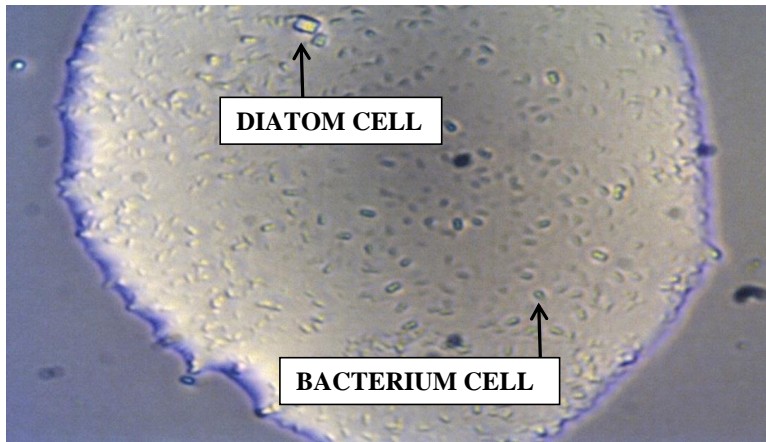

**Figure 3.** An example of bio-cargo incorporated into the jet droplet. Image taken with Zeiss
AxioCam ERc5s using Nikon Eclipse TS100 with 20X lens.

Bubbles rising in more concentrated diatom suspensions produced fewer jet droplets,

only ~5 from a diatom suspension of 11000 cells / ml, as compared with ~7 jet droplets
ejected from suspension of 3000 cells / ml (Table 1). This result indicates that the rising
bubble has a limited ability to collect diatom cells. Steadily accumulated bio-cargo reduces
also the speed of bubble rise and in turn decreases the energy available for upward projection
of bottom bubble vortex (see Fig. 4, Phase B). That is consistent with observations reported
by (Woolf and Thorpe, 1991), who observed that overloaded, so called "dirty bubbles" of 0.2
mm diameter rises with velocity of 0.6 cm/s as compared with "clean bubbles" of the same
diameter that surface with velocity 2.1 cm/s. A similar result was reported by Marks et al.
(2001) and showed that aerosolization of bacteria from polluted coastal sea water was
suppressed.

The rising bubble and induced diatom scavenge and related aerosolization is shown in

Figure 4. Four phases may be distinguished: separation of ions into cations and anions (Phase
A); scavenge of negatively charged cells from the water column (Phase B); sudden bubble
stop at the water/air interface and projection of cationic jet column that skims negatively
charged cells (Phase C); ejection of one jet droplet that contains diatoms and several without
bio-cargo along with spread of film droplets (Phase D). Distinction between the initial jet



droplets and secondary droplets were done by considering ejection heights according to
(Blanchard and Syzdek, 1982; Blanchard, 1989). During the experiments ejection height of
initial jet droplets was about 16-18 cm, while the secondary jet droplets were typically
projected to 3-12 cm (Table 1).
The enrichment factor, calculated as the number of diatoms included in the initial or
sub-initial jet droplet of volume $9.05 \cdot 10^{-6}$ ml divided by the number of diatoms suspended in
solution, indicated that diatoms incorporated into enriched jet droplets were directly derived
from the bubble bottom layer. Considering that the initial jet droplets also contains an
enhanced share of positively charged cations the process of diatom scavenge in the water
column and related ejection into air may operate on the electrostatic (cationic) adhesion of
negatively charged diatoms. Thus when bubble burst at the air-water interface diatoms could
be accommodated to a bit slowed (sub-initial) droplet (Fig. 4).

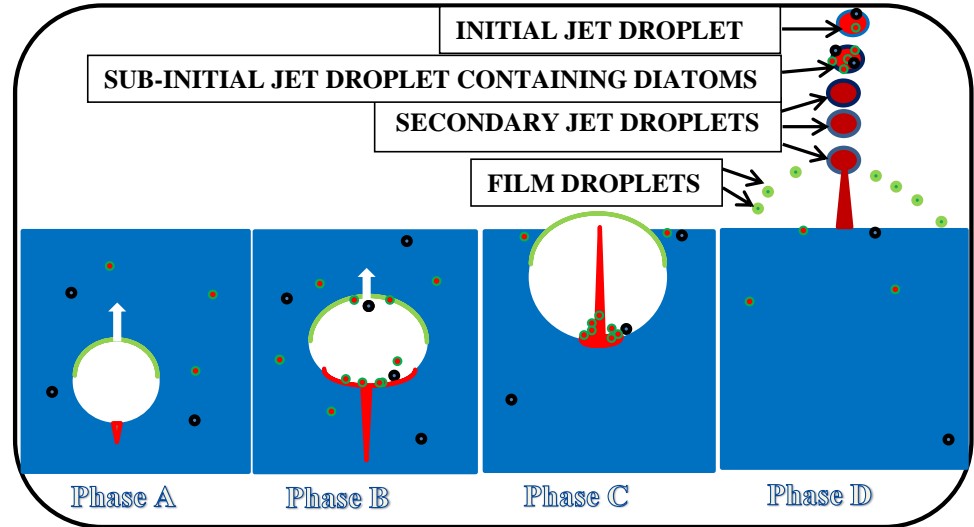

**Figure 4.** Four phases of a rising bubble mediated scavenge of bio-cells and related
aerosolization from: initial stage of developing upper anionic domain (marked by grey) and
bottom cationic vortex (marked by black) (Phase A); followed by cationic attraction and
scavenge of negative charged cells (Phase B); via projection of cationic jet, skimming
collected bio-cells (Phase C); to final ejection of initial and sub-initial jet droplets with bio-
cargo and secondary jet droplets along with spread of film droplets into the air (Phase D).

Assembled details regarding experimental conditions, including diatom shape, cell
concentrations, water temperature, salinity and estimated *Ef* for both investigated diatom
stains are listed in Table1. These observations relate to the bubble-mediated diatoms scavenge
suspended in two samples of different salinities, that are typical for brackish 15 g/kg and



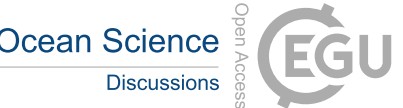

oceanic 35 g/kg salinities. Therefore the diatom stains were prepared to obtain typical natural
concentrations of diatom in sea water. The experiments were performed in solutions
containing 11000 *Nanofrustulum* and 3000 of C*yclotella* cells in 1 ml volume.

**Table 1.** Data set describing laboratory experiment evaluating diatoms aerosolization.

| MEDIUM/parameter | *Nanofrustulum* | *Cyclotella* |
|---|---|---|
| **DIATOMS/** | | |
| concentration (1 ml) | 11000 | 3000 |
| shape | elongated | spherical |
| size (μm) | ~3x12 | ~10 |
| detected charge on outer | negative | negative |
| **BUBBLE/** | | |
| ($D$) diameter (mm) | 1.2 | 1.2 |
| volume (ml) | ~$0.905 \cdot 10^{-3}$ | ~$0.905 \cdot 10^{-3}$ |
| detected charge (vortex) | positive (strong) | positive (strong) |
| detected charge (upper) | negative (weak) | negative (weak) |
| rise distance (mm) | ~100 | ~100 |
| **ENRICHED INITIAL OR SUB-INITIAL JET DROPLET/** | | |
| ($d$) diameter (mm) | ~0.12 | ~0.12 |
| volume (ml) | ~$0.905 \cdot 10^{-6}$ | ~$0.905 \cdot 10^{-6}$ |
| ejection height (cm) | 12-16 | 12-16 |
| detected charge | positive (strong) | positive (strong) |
| number of diatoms in enriched droplets | 1-12 | 1-3 |
| ($Ef$) enrichment factor | 370 (averaged) | 101 (averaged) |
| **SECONDARY JET DROPLETS/** | | |
| ($d$) diameter (mm) | ~0.12 | ~0.12 |
| volume (ml) | ~$0.905 \cdot 10^{-6}$ | ~$0.905 \cdot 10^{-6}$ |
| ejection height (cm) | 3-12 | 3-12 |
| number of droplets | 4 | 6 |
| detected charge | positive (weak) | positive (weak) |
| ($Ef$) enrichment factor | 0 | 0 |
| **WATER SOLUTION/** | | |
| ($Tw$) temperature (°C) | 22.1 | 22.1 |
| ($S$) salinity (g/kg) | 35 | 15 |


After experiments with diatom aerosolization from moderate concentrated suspension
of 11000 *Nanofrustulum* cells in 1 ml a next set with significantly concentrated medium of
840000 cells / 1 ml was completed. The conditions simulate a diatom bloom and refer to
salinity 35 g/kg. Obtained results showed much lower *Ef* values that ranged from ~5 to 30 and
steadily increased during the  elapsed time of experiment. The averaged value of *Ef* calculated
from all 20 samples was ~8 (Table 2).
Both experiments indicated that the averaged values of *Ef* estimated for diatoms
aerosolized form relatively low concentrated suspension was 101 for spherically shaped





*Cyclotella* cells and 307 for elongated *Nanofrustulum* cells (Table 1). However, the *Ef*
significantly dropped to 8 when *Nanofrustulum* cells were aerosolized from more
concentrated suspension (Table 2).

**Table 2.** Enrichment factor for *Nanofrustulum* aerosolized by uniform bubbles of D=1.2 mm
rising in water suspension of 35 g/kg salinity for distinctly different diatom concentrations.

| Parameter | | |
|---|---|---|
| Concentration (1 ml) | 11000 | 840000 |
| number of diatoms in enriched droplets | 1-12 | 1-23 |
| (*Ef*) enrichment factor | 370 (averaged) | 8 (averaged) |
| (*Tw*) water temperature (°C) | 22.1 | 22.1 |
| (*S*) water salinity (g/kg) | 35 | 35 |

**4  Discussion**

The collected evidences suggests that the bubble-mediated mechanism of diatom scavenge in
the water column and enrichment in aerosol droplets may depend on a combination of factors,
controlled by the positively charged (cationic) sub-bubble vortex (Marks, 2014) that attracts
negatively charged diatoms in the water. The sub-bubble vortex gathers rotational momentum
during the bubble rise in the water column and projects a small whirling water jet producing a
few jet droplets (Blanchard and Woodcock, 1957). If that concept is correct, the enrichment
factors of aerosolized diatoms and bio-cells may depend on the strength of bottom bubble
cationic vortex that forms a rotating pocket as well as the negative charge imparted to the
diatoms (this paper) or bacteria outer membranes (Blanchard and Syzdek, 1978; 1982; Weber
et al., 1983; Marks et al., 2001; Aller et al., 2005; Mayol et al., 2014).
The diatom uplift may be reduced as compared with bacteria (Blanchard and Syzdek,
1978) due to the higher weight of diatom cells. Thus the accommodation of diatoms into sub-
initial droplets derived from the water jet/column is more probable as depicted in Figure 4,
Phase C. Diatoms as relatively large objects suspended in the water thus are more resistant to
bubble-mediated drawing and related aerosolization, as compared with bacteria.
Extended laboratory experiment reported by Blanchard and Syzdek (1970) evaluated
the enrichment factor *Ef* of bacteria *Serratia marcescence* ejected with the initial jet droplets,
which exceeded 600, when the rise distance of bubbles through the bacteria suspension was c.
10 cm. Later, (Blanchard and Syzdek, 1978) determined that the concentration of bacteria is
always highest in the initial jet droplet and decreases progressively in the lower drops, being





lowest in the last ejected droplet. That however was not the case with ejection of diatoms.
Thus, at this stage of research we may only state that a single bubble at the moment of burst
projects diatoms which are incorporated to one of five-seven jet droplets.
The averaged values of *Ef* estimated for spherical *Cyclotella* cells was about 3-4 times
lower than that for the elongated *Nanofrustulum* cells. This indicates that the elongated cells
may integrate more negative anions on more expanded outer membranes, and these are more
effectively scavenged and aerosolized by bubbles. The *Ef* values obtained for diatoms exceed
two orders of magnitude the original concentrations in water suspensions, which indicate that
even relatively large and heavy diatoms, as compared with bacteria, are scavenged by rising
bubbles and aerosolized. Note that *Ef* values obtained in our laboratory study for aerosolized
diatoms are about 1-2 orders of magnitude lower than that obtained for bacteria (Blanchard
and Syzdek 1978; 1979; Marks et al., 2001; Mayol et al., 2014). However the values of *Ef*
decreased to 8 for the more concentrated suspension show that the efficiency of bubble-
mediated cells scavenge and aerosolization decreases with increasing content of cells
dispersed in the water. The range of *Ef* values obtained for diatoms is consistent with that
obtained for bacteria aerosolized from polluted sea reported by Marks et al. (2001).
The expanded interfacial of diatom cells, may accommodate relatively more negative
charge and contribute to enhanced cationic charge induced attraction of diatoms to the rising
bubbles. In contrast, the bacteria in aquatic systems are much smaller and more abundant, thus
bacterial concentrations in aerosols and sea surface microlayer are typically significantly
higher (Aller et al., 2005; Mayol et al., 2014) as compared with diatoms. However, the
charge around (in the outer membranes) and inside (most likely centered in the RNA/DNA
nucleus) the diatoms and bacteria cells implies that the cationic electrostriction is a key
enforcing factor on the overall distribution of electric charge in bio-cells.
The complementary experiments using a negatively charged Plexiglas exposed near
the place of bubble burst allowed observation of the trajectory of jet droplets which indicated
that all  jet droplets carry strong cationic loads. These observations also indicate that both
initial and sub-initial jet droplets were somehow more vigorously deflected towards
negatively charged Plexiglas plate as compared with secondary jet droplets (see Fig. 5).
Intriguing is also that enriched initial jet droplets, beside the condensed bio-cargo, also
incorporate a load of spiraling chains of cations (Marks, 2015). Observations indicate that
cations are gaining rotational energy under the pirouette narrowing of motion proceeded
within the sub-bubble vortex (Marks, 2015). That accelerated motion is then projected upward
when bubble bursts at the water surface and share of rotational energy may later contribute to




creation and accumulation of RNA/DNA inside airborne droplets (Marks, 2015). In addition,
the highly energetic cationic vorticity may permeate into just collected bacteria or diatom
cells, trapped inside bio-droplet, as generally illustrated in Figure 5.

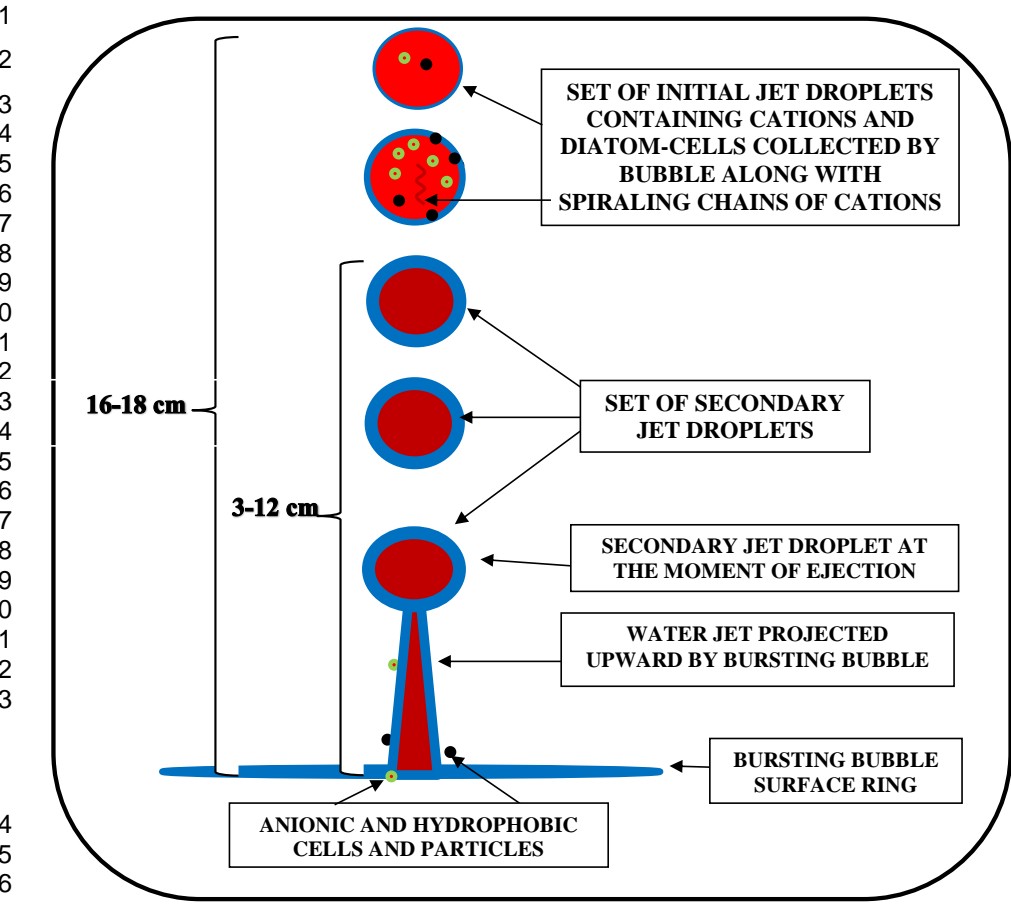

**Figure 5.** Illustration of initial and secondary droplets ejecta that results from bubble burst at
the water surface (see Fig. 4, Phase D) projecting jet droplets including cationic initial jet
droplets skimming negatively charged diatoms.

Distinction between the initial and secondary jet droplets were done by considering
ejection heights according to Blanchard and Syzdek (1982). During experiments in the present
study, the ejection height of initial jet droplets was about 16-18 cm, while the secondary
droplets were typically projected to 3-12 cm (Table 1, Fig. 5).
Qualitative measurements of electric charge fluctuations accumulated on the rising
bubble outer (Table 1) conducted by using oscilloscope indicated a high fluctuation of charge
between bulk water and bubble boundaries, when bubble collided with probe head placed at



water surface. The charge polarity, indicated positive impulses that were roughly about 4
times stronger (due to converging thus more condensed cationic domains) compared to the
negative impulses due to diverging thus les condensed anionic domains as illustrated in Fig. 6.
In addition, the negative net charge of *Nanofrustulum* community (Table 1) adhered into the
oscilloscope probe was confirmed.

Collected evidences shows that the cation-mediated electrostriction plays a principal

role in bubble vorticity and related attraction/scavenge of bacteria and diatoms by bubbles
(Marks, 2014). The efficiency of diatoms scavenge may also depend on anionic charge
gathered on outer membranes (Gottenbos et al., 2001) which may depend on more individual
bio-cell properties, perhaps related to the charge incorporated into RNA/DNA nucleus. A
similar process of cation-induced gathering of anions may stimulate development of negative
outer membrane in bacteria (Gottenbos et al., 2001).

In order to underline the rising bubble-induced cationic dominance, one photographic

image, taken during the experiment tracing bubble rotary motion in a clean (filtrated) sea
water temperature of $38.0^o$C is shown in Figure 6. The image shows three tracers, two of
which (a and b) depict fast rising bubbles sustaining only the sub-bubble cationic rotaries
(presumably the anionic rotary was transferred off the bubble). The captured moment, shows
two sub-bubble cationic rotaries that interact with each other. A longer tracer of faster rising
bubble (a) is passing a slower tracer (b). Both tracers show levorotary spiraling in upward
directed motion. The captured tracers deflects each other due to the cationic electric charge, to
prevent otherwise inevitable coalescence. In addition, a sinistral (counterclockwise) whirling
of both upward directed tracers is visible revealing a strong, converging sub-bubble cationic
motion (Fig. 6). To contrast that rather unusual case also a more typical rising bubble tracer
(c) is shown revealing a case when both anionic and cationic vorticities are sustained around
the bubble, see tracer (c) in Fig. 6.

The illustrated case shows that bubbles in clean and warm sea water may rise

relatively faster (Fig. 6, tracers a and b). Thus, such rising bubbles may disperse the anionic
rotary outward, and the internal cationic vorticity can be visible. The photograph reveals that
convergence and related lining-up of cations, ongoing in the rising sub-bubble wake is a
dominating rotational feature, while the anionic rotary is weaker, perhaps aligned feature.

In open water, which contains substantial loads of suspended matter (including viable

bacteria and diatoms) the bubble rotational features may be altered. The velocity of bubble
rise may be reduced depending on the quantity and size of particulates collected by bubbles.



From time to time, bubbles may become overloaded with respect of collected cargo that may
be shed and disintegrate upper and bottom bubble rotaries.

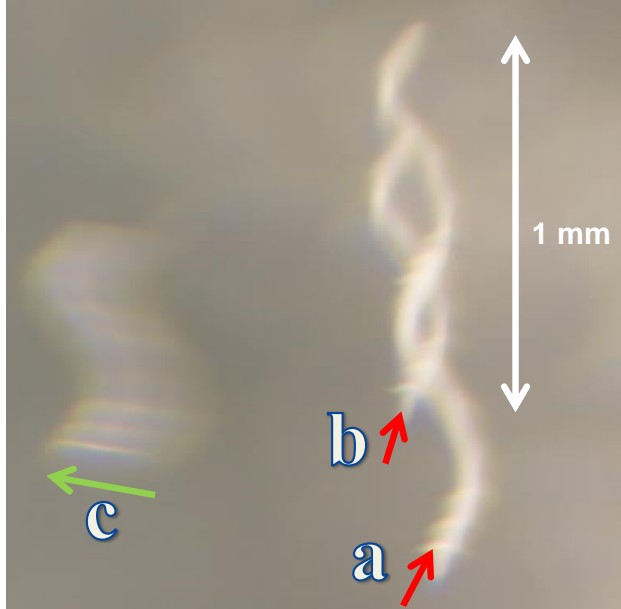

**Figure 6.** Rising bubble cationic-rotary tracers (a, b) and anionic/cationic tracer (c) assembled
in filtrated, clean sea water taken from Pomeranian Bay at $Tw = 38.0^{\circ}$C and $S = 8$ g/kg.
Bubbles were produced in laboratory conditions by fuzzy salt (Sal EMS Factitium); image
was taken by Tamron Xr Di 28-75 mm lens with reverse ring in 1/10 s time.
The outlined cationic mechanism seem to play a principal role in organizing motion of

ions around bubbles in saline water. The mechanism include also a steady assembly of
rotating cations that are ejected into the droplets enriched by preselected biota. Thus the same
(coherent) bubble-cationic-rotational processing of matter ongoing in the ocean may assemble
coherent bio-active (cation-active) molecules that formed diverse biota.

**5      Conclusions**

Experiments show that rising bubbles in saline water can develop a strong cationic vortex that
scavenge diatoms in water, which are transported to the surface and eject as jet droplets into
air. The mechanism operates on cationic principles that include:

1) effective cationic-mediated scavenge of negatively charged diatoms that are

collected in large numbers during bubble rise in the water column,



2) diatoms are collected and concentrated in rotating bubble bottom pocket,
3) diatoms are aerosolized in sub-initial jet droplets when bubble bursts.
The enrichment factor for aerosolized diatoms strongly decreases with increasing
concentration of suspended matter in the water column.
In addition, experiments indicate that the cationic selection of diatoms depend on the
cells outer negative charge that may enhance bubble-mediated scavenge and aerosolization.
Bubble-mediated cationic scavenge of bio-cells in clean saline water and related
aerosolization may contribute to global bio-matter cycling and related process of matter
accumulation near the ocean surface. Thus massive and long-term bubble-cationic-rotational
processing of matter in the oceanic water and in droplets suspended in the troposphere may
likely incepted the bio-matter evolution on the Earth.

*Acknowledgements.*Diatom strains have been isolated within the frame of Maestro Research
Grant 2012/04/A/ST10/00544 funded by National Science Centre in Cracow, Poland.

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
