# Peer review of "Rising bubbles as mechanism for scavenging and aerosolization of diatoms 2 Roman Marks1, Ewa Górecka2 Kevin Mc Cartney3, Wojciech Borkowski1 3 4 5 1 University of Szczecin, Faculty of Geosciences, Physical Oceanography Unit, 6 Mickiewicza"

_Ocean Science, 2017_

## Referee Comment (RC1) · Anonymous Referee #1 · 5 Dec 2017

The manuscript entitled "Rising bubbles as mechanism for scavenging and aerosolization of diatoms" discusses experimental studies of the enrichment of diatoms in jet droplets produced by bursting bubbles in laboratory experiments. The basic finding of the study indicates that diatoms are highly enriched in bubble bursting aerosol (also called sea spray aerosol). This simple finding is reasonable and aligns with prior studies of similar marine microbiota, although this reviewer finds various problems with the mechanistic explanations, including various possible simpler explanations that are equally valid based on what is known about bursting bubbles in the literature, and are noted in the comments below.

Experimental data are not presented in the manuscript, except for certain broad statistics in basic tables – some of which are highly qualitative. The figures entirely consist

[Figure]

of cartoons or exemplary images. The findings of this study should be provided in such a way that the reader (and reviewer) can evaluate the data, assess its quality, and use the manuscript as an aid when attempting to reproduce the findings at some time in the future. Based on the current form of the manuscript, this is not possible, and the manuscript should certainly not be accepted for publication until this is corrected.

It is recommended that this manuscript should be considered for publication in Ocean Science only after very major revisions are made to nearly all aspects of the paper.

The introduction could be substantially simplified. It seems to ramble about various aspects of bubble physics, not all of which are related to the focused finding of this paper. It is the recommendation of this reviewer that substantial revisions be made to this section overall.

Line 102: "...no bubble related research has focused on diatom scavenging [sic]..." Atmospheric chemists have been increasingly interested in this topic over the past 5-10 years. Images of phytoplankton and their fragments in sea spray aerosol samples can be found in the literature (e.g., Bigg and Leck, Tellus B 2005; Patterson et al., ACS Cent. Sci. 2016; Lee et al., J. Phys. Chem A 2015).

The initial/sub-initial droplets generated in this study are very large and will sediment quickly, rendering their atmospheric relevance to be minimal. Table 1 shows that the secondary drops are also d = 0.12 mm. Were these measured or deduced based on bubble diameter? In a recent study (Wang et al., Proc. Nat. Acad. Sci. 2017), secondary droplets were proposed to be smaller than the initial/sub-initial jet droplets. The aforementioned study by Wang et al. also provides a detailed discussion of sea spray production and its charge distribution from an aerosol perspective, and may be of general utility in the discussion of the results of the present study.

Line 191: "Availability of cations decreases during aeration" Please provide a citation for this assertion.

[Figure]

Line 198: Please expand on the manner in which the electric charge distribution around rising bubbles were collected. The explanation is inadequate to allow a reader to reproduce the measurement.

Line 237: "Bubbles rising in more concentrated diatom suspensions produced fewer jet droplets" The conclusion given is that the rising bubble had a limited ability to interact with the diatoms. The problem with this argument is that the observation in question is a change in total jet droplet production, not diatom-included droplets. A reduction in diatom inclusion may not drive a reduction in droplet production – but the converse may be true! A simpler, and equally valid, explanation is that jet droplet production was suppressed by a change in the composition of the water (and likely its physicochemical properties, like surface tension and viscosity), which also changed the mechanics of diatom inclusion in the initial/sub-initial droplet.

Line 240: It is difficult to conceptualize the relationship between bubble rise speed and the kinetic energy of the aerosol resulting from the bursting event. These are more likely separate processes, where the kinetic energy in the bubble rise is dissipated into the interfacial tension of the fluid at the air sea interface, and/or translational/rotational degrees of freedom of the bubble itself that are not related to the fluid fragmentation process of the bubble bursting event. Bubbles may reside at the water surface for a time before bursting, for lengths of time that depend on various parameters. The authors' explanation is difficult to verify based on the given information. In addition, the ability of a bubble to eject a cell is likely not limited by the energy required for ejection. The density of a cell is very close to that of the surrounding seawater (while it is acknowledged that cells are known to adjust their density to control buoyancy). The energy required to eject a diatom-free droplet is therefore likely very similar to the energy required to eject a diatom-laden droplet, although physical quantities to describe this difference are not known to this reviewer.

Suppression of jet droplet production has been shown in seawater enriched in diatoms and their associated dissolved organic matter when foams form on the water surface
(e.g., King et al., Environ. Sci. Technol. 2013; Collins et al., Atmos Meas Tech 2014). It is important to consider the diatoms while including the dissolved organic carbon that they produce routinely. This comment also relates to the discussion on lines 328-338.

Figure 4: The conceptual framework of this figure is confusing. Consideration is not given to the hydrodynamic features of the diatoms themselves, yet such consideration is given to the much smaller ionic species. An explanation of diatoms existing at the base of a droplet may be associated with hydrodynamic drag during bubble ascent, where the larger particles migrate to the base of the bubble. More importantly, Phase C seems to suggest that the bubble bursting event is driven by the production of a cationic jet prior to film cap rupture. Such a concept is inconsistent with the known mechanism for bubble bursting. The detailed treatment by Lhuisseur and Villermaux (J. Fluid. Mech. 2012) [along with others prior and since] shows that the formation of the jet is related to the collapse of the bubble cavity following film cap rupture. This figure and associated conceptual model is therefore extremely confusing at best and may include fundamental flaws.

Table 1: If the initial/sub-initial droplets and secondary droplets are distinguished by ejection height, how was a lack of enrichment factor derived for secondary droplets. Any collections at low ejection heights would necessarily require the collection of initial/sub-initial droplets as well by consequence. Please clarify the method used to derive these quantities and provide data in plots and/or in entirety as supplemental information.

Table 2 shows that the range of the number of diatoms existing in enriched droplets widens by a factor of 2 for the more concentrated sample. A listing of the statistical parameters of the distribution of observations from these experiments would substantially improve the clarity of these results in the context of the discussion within the text. Providing a histogram of observations (e.g., a frequency distribution of observing 'N diatoms per droplet') would help mightily.

Line 357-363: This conceptual framework (paired with Figure 4) may be inconsistent with the current paradigm based on detailed observations by multiple investigators – or the explanation may be unclear. A greater defense of the proposed mechanism should be provided and/or the authors' argument should be presented in light of the current evidence-based understanding of the bubble bursting mechanism in which the basal jet is formed during bubble cavity collapse, and not associated with the bubble stopping as it approaches the (bulk) air-water interface.

Line 365: In order to support this argument, please cite evidence to suggest that diatom cells are more dense than seawater and bacteria cells.

Line 367: Diatoms are indeed larger than bacteria, and therefore are more subjected to hydrodynamic drag as bubbles ascend through the water column. This simple argument (as discussed above) would provide an equally valid case for inclusion of diatoms in the initial/sub-initial droplets that are derived from the jet. It is important not to forget that the concentration of bacteria in the ocean also dwarfs the number concentration of diatoms by multiple orders of magnitude, so the statistical likelihood of finding a bacterium in a droplet is simply greater.

Line 374-376: Can the data that support this statement be shown in the manuscript or the supplement?

Line 403-410: This discussion is off topic and confusing. Define a 'spiraling chain of cations'. What sort of cations are involved? This is particularly hard to envision with inorganic ions. How is this related to the topic of the manuscript? Where are the DNA and RNA bases coming from? This paragraph could be easily omitted without a loss of context or substantive discussion of the results. In general, the discussion of the cationic vortex and related items are largely irrelevant to the stated motivation of the study. The discussion of this topic is also confusing and not well buttressed by evidence. It is interesting to think about the charge separation that reportedly occurs in the rising bubble, but the deeper mechanistic discussion strays from the important

point.

Line 446-453: Much greater detail on the methods and findings (including visualized data!) that reflect the description in this paragraph should be included in the manuscript.

Line 507: The conclusions relating to differences in concentration should be tempered, as the mechanism may not be well explained or explored by this study, and only two concentrations were attempted.

Line 513: The claim in the last sentence is almost entirely unrelated to the study at hand, and at the same time, is a significant stretch in logic. Prebiotic chemistry is, indeed, an exciting field, but the simple fact that a vortex exists beneath a rising bubble does not necessarily suggest that this is related to proto-nucleic acid formation – although in making this argument, it is clear to see that this subject of the detailed structure of cationic vortexes is quite off topic for the current manuscript.

Other comments: Line 132: "bubble mediated scavenge [sic] and aerosolization of biomolecules with special attention given to diatoms" The authors perhaps mean to say 'aerosolization of biomaterials' as diatoms are not cells and not molecules.

The proper terminology for diatom morphology should be used: Cyclotella are 'centric' diatoms, while Nanofrustulum are 'pennate' diatoms.

Diatoms have been referred to in this manuscript in general terms using "bio-cells" or "bio-matter". These are not accepted terms and the authors should simply refer to them as "cells", "biological material", or even "biological particles", as are often used in the oceanography literature when referring to filter-retained material in the ocean.

Concentrations should be provided using standard unit notation (cells/mL) rather than noted as "cells / 1 mL." In tables, units of volume (mL) are consistently given instead of units of concentration. It is suggested that the authors seek further editing of English usage and grammar for this manuscript.

---

## Referee Comment (RC2) · Anonymous Referee #2 · 8 Dec 2017

The manuscript presents some interesting postulations on the mechanisms for the scavenging of diatomaceous material by rising bubbles and the aerosolisation of scavenged material on bubble-bursting at the sea-air interface. I am reviewing this from the perspective of marine atmospheric science, where there is a substantial body of literature concerned with the bubble-mediated production of seaspray aerosol. Convergence of interpretations of evidence in this field will be extremely important in resolving outstanding mechanistic uncertainties and the current work aims to provide mechanistic insight into this debate. However, as it is written, I have some general problems in understanding the concepts and implications of this work and there are quite a few points that must be addressed before the manuscript is publishable.

Context: First, I have a couple of points about the context of the work. Much atmo-

spheric work has focussed on the enrichment of organic material in seaspray particles. Owing to the much greater number of particles in sizes from around one to a few hundred nm and the importance of these as potential cloud nuclei, much of the work has concerned characterisation of organic material in these particles in terms of its broad chemical functionality and influence on water uptake in the moist atmosphere (O'Dowd et al., 2004; Wang et al., 2017). As recognised at the top of section 1.3 in the current manuscript, primary marine particles in these size ranges are from the "film" mode, formed when the bubble cap shatters on bursting, not from the larger sized, but less abundant, "jet" mode of particles. It has been shown that enrichment of diatomaceous exudate in this film mode is of more atmospheric significance than that of organic material of other phytoplankton origin (Fuentes et al. 2010), but there has been less recent atmospheric focus on aerosolised intact diatoms since they will generally be too large to be found in the abundant film particle mode and will not thereby influence liquid cloud droplets. However, there has been recent interest in the potential roles of diatoms as the much scarcer ice nucleating particles (Wilson et al., 2015). For this reason (and for mass budgeting and biogeochemical cycling purposes, as recognised in the few recent studies in this area), mechanisms for their atmospheric aerosolisation will be important (particularly under low wind speed conditions relevant for the Arctic atmosphere). I believe these contextual considerations are important to set the scene for the current work. However, once the atmospheric context is appreciated, the relevance of enrichment of diatoms in droplets of 120 microns diameter becomes apparent and the likelihood of such enrichments playing a role on atmospherically important timescales must be seen as negligible.

Abstract: It is not clear to me that the abstract serves the usual purpose of an abstract. Almost all the text is background information that does not summarise the findings of the paper and refers to previous literature (mainly from the first author). Indeed, the only part of the abstract that relates specifically to the paper findings is the last sentence.

Introduction: I found this rather meandering and not all directly relevant to the current work. The last paragraph of section 1.3 outlines the findings of previous experiments on the dynamics of bacterial scavenging by rising bubbles and states that the novelty of the current study is "...the bubble mediated scavenge [sic] and aerosolization of bio-molecules with special attention given to diatoms".

Methodology: I have a number of queries about the experimental procedure. i) what is the air pump flow rate? This would be useful to understand the scavenging dynamics. 1 sccm would be $1.667 \times 10^{-5}$ L/s. Each 1.2 mm diameter bubble is $0.905 \times 10^{-6}$ L, so this would be 18.42 bubbles per second (@ 1 sccm), so 3 seconds of bubbles with 7 jet drops per bubble would be 387 droplets collected at this flow rate. At a droplet diameter of 0.12 mm, this is $3.5 \times 10^{-7}$ L = 0.35 microlitre. ii) how was the bubble size measured (optically or acoustically)? iii) how was the droplet size measured? iv) was the RH actively controlled to ensure the water evaporation rate was constant for all experiments and hence the diatom concentration was being measured at the same water activity (and hence normalised to ionic strength)

Given the 150 ml volume, I do not understand how sampling roughly of the order of 1/3 microlitre in 3 seconds can substantially lead to "both diatoms concentrations in the water suspension and water salinity (or more precisely the availability of cations) decrease during aeration". Obviously the air pump flow rate could be very much higher, but then this would lead to many thousands of droplets on the slides.

Consequently, I am not sure of the statement on lines 194-196: "after analyzing that set of data we noticed that the efficiency of diatom aerosolization increased, indicating that rising bubbles reduced concentrations of diatoms (solution was gradually cleaned) during the experiments."

Results: My most significant concern is with the presentation of the results - or more precisely, the lack of systematic presentation of results. The results should be detailed along with errors and statistical distributions to support the hypotheses. The descriptions of the results is hard to follow as it is currently written, since it is frequently mixed with discussion material (e.g. lines 219-222, 230-232, 240-247). When this discussion is removed, it become apparent that there are few results presented. Table 1 refers to very few droplets, from which I cannot understand the methodology (number of repeats, flow rates and sampling strategy etc...) Moreover, many of the the discussion points are either incorrect of logical non-sequiturs (e.g. line 239 - there is no reason for an inability of bubbles to scavenge diatoms to lead to a reduction in jet droplets and line 258 - there is no reason that an enrichment factor indicates that diatoms were from the bubble bottom layer, though they may have been).

I am not sure that I follow the attempt to distinguish the initial and secondary droplets through the ejection heights as outlined in lines 253-257 when it is clearly stated that there was "not exclusively top jet droplets were enriched by diatoms" on 219 and "Conducted screening showed that only 20-25% of jet droplets were enriched by diatoms, which suggests that the process of diatom aerosolization might be influenced by a combination of factors operative in the water column, or at air-water interface". I would have liked to have seen some statistical analysis of the distribution of enrichments with height (for example) to support the mechanistic contentions and postulations.

Discussion & Conclusions: The discussions section presents some interesting conjecture, but does little to draw on and add to the results of the paper. Furthermore, I do not believe that the authors have presented results that can be evaluated sufficiently rigorously to support the stated conclusions.

Minor The sentence starting on line 48 reads peculiarly. The statement "generated at wind velocity 8 m/s" makes the sentence appear to be a non-sequitur, since the previous sentence states simply that the lume depth is the same as the wave height (with no wind speed dependence). If this is simply additional information, the sentence can straightforwardly be rephrased. The second phrase needs further punctuation "while, during rain,"

line 66 "Cl/Na atomic mass..." add "RATIO"

O'Dowd, C. D., Facchini, M. C., Cavalli, F., Ceburnis, D., Mircea, M., Decesari, S., Fuzzi, S., Yoon, Y. J., and Putaud, J.-P.: Bio- genically driven organic contribution to marine aerosol, Nature, 431(7009), 676–680, 2004

Wang, X., G. B. Deane, K. A. Moore, O. S. Ryder, M. D. Stokes, C. M. Beall, D. B. Collins, M. V. Santander, S. M. Burrows, C. M. Sultana and K. A. Prather: The role of jet and film drops in controlling the mixing state of submicron sea spray aerosol particles, PNAS 114 (27), 6978-6983, doi:10.1073/pnas.1702420114, 2017

Fuentes, E., Coe, H., Green, D., de Leeuw, G., and McFiggans, G.: On the impacts of phytoplankton-derived organic matter on the properties of the primary marine aerosol – Part 1: Source fluxes, Atmos. Chem. Phys., 10, 9295-9317, https://doi.org/10.5194/acp-10-9295-2010, 2010

Wilson, T. W., Ladino, L. A., Alpert, P. A., Breckels, M. N. B., Brooks, I. M., Browse, J., Burrows, S. M., Carslaw, K. S., Huffman, J. A., Judd, C., Kilthau, W. P., Mason, R. H., McFiggans, G., Miller, L. A., Nájera, J. J., Polishchuk, E., Rae, S., Schiller, C. L., Si, M., Temprado, J. V., Whale, T. F., Wong, J. P. S., Wurl, O., Jakobi-Hancock, J. D. Y., Abbatt, J. P. D., Aller, J. Y., Bertram, A. K., Knopf, D. A. K., and Murray, B. J.: A marine biogenic source of atmospheric ice-nucleating particles, Nature, 525, 234–238, https://doi.org/10.1038/nature14986, 2015

---

## Author Comment (AC1) · 18 Jan 2018

Refer to: Ocean Science: os-2017-82, submitted on 03 Oct 2017 Manuscript Number: PAAG-D-14-00412R2; Title: Rising bubbles as mechanism for scavenging and aerosolization of diatoms; Authors: Roman Marks, Ewa Górecka, Kevin Mc Cartney, and Wojciech Borkowski Responses to the Anonymous Referee #1

First of all I would like to thank the Reviewer for very helpful comments. I have considered majority of them and accordingly altered the manuscript.

1. Comment The manuscript entitled "Rising bubbles as mechanism for scavenging and aerosolization of diatoms" discusses experimental studies of the enrichment of diatoms in jet droplets produced by bursting bubbles in laboratory experiments. The

basic finding of the study indicates that diatoms are highly enriched in bubble bursting aerosol (also called sea spray aerosol). This simple finding is reasonable and aligns with prior studies of similar marine microbiota, although this reviewer finds various problems with the mechanistic explanations, including various possible simpler explanations that are equally valid based on what is known about bursting bubbles in the literature, and are noted in the comments below. Experimental data are not presented in the manuscript, except for certain broad statistics in basic tables – some of which are highly qualitative. The figures entirely consist of cartoons or exemplary images. The findings of this study should be provided in such a way that the reader (and reviewer) can evaluate the data, assess its quality, and use the manuscript as an aid when attempting to reproduce the findings at some time in the future. Based on the current form of the manuscript, this is not possible, and the manuscript should certainly not be accepted for publication until this is corrected. It is recommended that this manuscript should be considered for publication in Ocean Science only after very major revisions are made to nearly all aspects of the paper. The introduction could be substantially simplified. It seems to ramble about various aspects of bubble physics, not all of which are related to the focused finding of this paper. It is the recommendation of this reviewer that substantial revisions be made to this section overall.

2. Response The approach to construct the experimental set-up was mechanistic; therefore also results and explanations are mechanistic. The motivation to undertake that research was a discovery that bubbles are assembling bi-ionic/bi-rotational features. The features (dominating cationic rotary, and weaker, probably aligned anionic one as well as even double helix like motion) was documented by tracing bubble photo-transcripts, and confirmed by complementary experiments (Marks, 2014; 2015). The best records were collected when cylinders with water/bubbles were lighted from below or top, and when water was clean (filtered) as well as relatively warm. Also preliminary experiments conducted in natural sea water show that bubbles develop rotary, although transcripts are rather very chaotic and fogy especially when water contains significant share of suspended materials. In addition, we proved that there are charge polarities

within the bubble upper and bottom half spheres (the positive charge of jet droplets was reported Blanchard (1963) and documented by means of experiments). That is why we used the mechanistic/electrostriction approach to explain the extreme efficiency of bubble-mediated biological particles scavenges. Since bacteria scavenge was reported by several researchers we conducted experiments that now allow estimation of enrichment factors Ef for aerosolized diatoms. Considering the content of comment the manuscript is rephrased and several new paragraphs as well as a new Fig. 5, showing data distribution is included along with additional information on number of samples collected during all sets of experiments is provided in Tables 1 and 2. Several new or altered sentences are inserted according to Reviewer specific comments.

3. Changes to manuscript (see specific comments)

Specific comments 1) Comment Line 102: ". . .no bubble related research has focused on diatom scavenging [sic]. . ." Atmospheric chemists have been increasingly interested in this topic over the past 5- 10 years. Images of phytoplankton and their fragments in sea spray aerosol samples can be found in the literature (e.g., Bigg and Leck, Tellus B 2005; Patterson et al., ACS Cent. Sci. 2016; Lee et al., J. Phys. Chem A 2015). The initial/sub-initial droplets generated in this study are very large and will sediment quickly, rendering their atmospheric relevance to be minimal. Table 1 shows that the secondary drops are also d = 0.12 mm. Were these measured or deduced based on bubble diameter? In a recent study (Wang et al., Proc. Nat. Acad. Sci. 2017), secondary droplets were proposed to be smaller than the initial/sub-initial jet droplets. The aforementioned study by Wang et al. also provides a detailed discussion of sea spray production and its charge distribution from an aerosol perspective, and may be of general utility in the discussion of the results of the present study

2. Response The abstract and introduction is shortened and reedited. A new paragraph indicating contribution of atmospheric researchers to study the composition of sea spray is inserted. The sizes of bubbles and initial/subinitial droplets were scaled during tests performed in laboratory conditions using photography method by tracing

bubbles transcripts and jet droplets transcripts just after injection into air. This way the general rule 1/10 was confirmed. The relative humidity in the laboratory room was much lower than 100%, but the experiments were conducted in semi closed glass breaker, under continuous aeration, thus the droplets collection was done under saturated or almost saturated humidity. After exposition microscopy slides were immediately placed and closed in the Petri Dish with small volume of water, to avoid damage of cells that were kept under air humidity of 100%. In addition, as suggested; Line 102 was altered. The design of experiment was not appropriate to study the size distribution of 1, 2, 3, 4 or 5 jet droplets in set. However, in literature, the jet droplet sizes in tens of mm size ranges were investigated by Blanchard and Syzdek 1975, Limnology and Oceanography, 20, 5, where in Fig. 4 a tendency of increase of droplet sizes in the jet set is plotted. Since our droplets were in the same range of sizes we are not referring to particular data reported by (Wang et al., 2017), who researcher sub-micrometer droplets produced by hydrogen bubbles generated through the electrolysis. Although we refer to that paper elsewhere in the altered manuscript when referring to cloud properties. Considering the Reviewer suggestion the explanation regarding the size of jet droplets was included to methods (second paragraph).

3. Changes to manuscript [Abstract. Bubbles rising in relatively clean saline water cause steady displacement of ions at the bubble boundaries which creates electric polarity that draw biological particles dispersed in the water. Viable diatoms as well as bacteria develop negative charge on outer membranes that are attracted to the cationic bubble bottom half sphere. When bubble bursts at the air/water interface the diatoms and bacteria are ejected into the air with initial or secondary jet droplets that are projected upward with a small water column derived from a sub-bubble cationic vortex. Laboratory experiments conducted in brackish and oceanic saline water on Nanofrustulum and Cyclotella cells indicate that the average concentration of jet droplets compared to original water volume (here termed the enrichment factor) for aerosolized diatoms may range from 8 to 307.]

[Increasing interest in biological and chemical nature of sea salt aerosols has resulted in numerous field experiments and papers e.g. ( Bigg and Leck 2005; Patterson et al., 2016). In particular the role of sea spray particles in determining electrical mobility of particles and related cloud properties was investigated by (Wang et al., 2017). In addition, evidences that aerosol samples contains phytoplankton and fragments of diatoms were reported by e.g. (Bigg and Leck, 2005; Lee et al., 2015; Patterson et al., 2016). Furthermore, the importance of organic materials as cloud condensation nuclei and broader atmospheric functionality is reviewed by (O'Dowd et al., 2014). Apparently, also the contribution of diatomaceous is of more atmospheric significance than the other organic materials of a phytoplankton origin (Fuetnes et al., 2010). It has been also suggested that fragments of diatoms may provide ice nucleating particles (Wilson et al., 2015). However, no particular investigations on enrichment of diatom cells in jet droplets have been reported.]

[….an experimental set-up was tested and sizes of rising bubbles and jet droplets tracers were scaled using photographic method (bubble tracers were captured during rise in the water column, while tracers of jet droplet were captured just after droplet ejection);].

List of literature was expanded and include:

[Bigg C., Leck E.K. (2005) Biogenic particles in the surface microlayer and overlaying atmosphere in the central Arctic Ocean during summer, Tellus B, 57,4, 305-316, DOI: 10.1111/j.1600-0889.2005.00148.x.

Lee C., Sultana C. M., Collins D. B., Santander M. V., Axson J. L., Malfatti F., Cornwell G. C., Grandquist J. R., Deane G. B., Stokes M. D., Azam F., Grassian V. H., Prather K. A. (2015) Advancing Model Systems for Fundamental Laboratory Studies of Sea Spray Aerosol Using the Microbial Loop. J. Phys. Chem. A, 119338860–7010.1021/acs.jpca.5b03488.

Lakhina G.S. (1993) Electrodynamic coupling between different regions of the atmosphere. Current Science, 64, 660-666.

O'Dowd C. D., Facchini M. C., Cavalli F., Ceburnis D., Mircea M., Decesari S., Fuzzi S., Yoon Y. J., and Putaud J.-P. (2004) Bio- genically driven organic contribution to marine aerosol, Nature, 431(7009), 676–680.

Patterson J. P., Collins D. B., Michaud J. M., Axson J. L., Sultana C. M., Moser T., ... Gianneschi N.C. (2016) Sea Spray Aerosol Structure and Composition Using Cryogenic Transmission Electron Microscopy. ACS Central Science, 2(1), 40–47. http://doi.org/10.1021/acscentsci.5b00344.

Wang X., Deane G.B, Moore K.A., Ryder O.S., Stokes M.D, Beall C.M., Collins D.B, Santander M.V., Burrows S.M., Sultana C.M. and Prather K.A. (2017) The role of jet and film drops in controlling the mixing state of submicron sea spray aerosol particles, PNAS 114 (27), 6978-6983, doi:10.1073/pnas.1702420114. Wilson T.W., Ladino L.A., Alpert P.A., Breckels M.N.B., Brooks I.M., Browse J., Burrows S.M., Carslaw K.S., Huffman J.A., Judd C., Kilthau W.P., Mason R.H., McFiggans G., Miller L.A., Nájera J.J., Polishchuk E., Rae S., Schiller C. L., Si M., Temprado J.V., Whale T.F., Wong J.P.S., Wurl O., Jakobi-Hancock J.D.Y., Abbatt J.P.D., Aller J.Y., Bertram A.K., Knopf D.A.K., and Murray B.J. (2015) A marine biogenic source of atmospheric ice-nucleating particles, Nature, 525, 234–238, https://doi.org/10.1038/nature14986.]

1) Line 191: "Availability of cations decreases during aeration" Please provide a citation for this assertion. 2) We are not in the position to provide a citation or exact values. We however infer that cations are continuously removed form a 150 ml of solution, during a few minutes of aeration. However, I agree that also the other conditions, and in particular related to air/water interface might change during aeration. Therefore the "availability of cations" was deleted and whole paragraph altered to read: 3) [Taking into account that during aeration the air/water interface develops a microlayer and that diatom concentrations in the water suspension decrease, only 8 samples were collected to estimate enrichment factor for aerosolized Cyclotella diatoms from suspension of low concentrated diatom cells. In case of high diatom concentration of Nanofrustulum, the number of

samples was extended to 20. However, after analyzing that set of data we noticed that the efficiency of diatom aerosolization decreased in time of aeration, indicating that rising bubbles reduced the concentrations of diatoms in the water penetrated by bubbles (solution was gradually cleaned). In addition, the development of microlayer and changes in physicochemical properties of the water/air interface gradually suppressed the bubble mediated diatoms aerosolization.] 1) Line 198: Please expand on the manner in which the electric charge distribution around rising bubbles were collected. The explanation is inadequate to allow a reader to reproduce the measurement.

2) In order to provide that information a following is inserted:

3) [In order to detect electric charge distribution around the rising bubble boundaries an oscilloscope, type HANTEK®-DS01201BV was used, allowing to trace the voltage polarity between a two probe heads in mV with 0.1 mV detection limit and 0.5% accuracy. One head was submerged to the depth of 2 cm in the bulk water, and the second probe tip was placed just at the water/air/bubble interface. The measurements were conducted in a small glass tank filed with 700 mL of saline (35 g/kg) water of 21.0oC temperature. Bubbles D = 1.2 mm were generated by capillary aerator with a rate allowing ca. 20 mm distance between a successive bubbles. These rose about 6 cm in a free water column and then were slowed on a glass plate tilted to 30o that reduced a zig-zag motion of bubbles, allowing interactions with the tip of probe head. The measurements showed negative charge introduced by bubbles outer layer that typically adhered to the tip at water/air/bubble interface. However, from time to time also strong positive charge was measured, indicating interaction with cationic domain gathered within surfacing bubble vortex. The maximum negative and positive values were considered as revealing polarity of ions gathered around the bubble upper and bottom half spheres.] 1) Line 237: "Bubbles rising in more concentrated diatom suspensions produced fewer jet droplets" The conclusion given is that the rising bubble had a limited ability to interact with the diatoms. The problem with this argument is that the observation in question is a change in total jet droplet production, not diatom-included droplets.

A reduction in diatom inclusion may not drive a reduction in droplet production – but the converse may be true! A simpler, and equally valid, explanation is that jet droplet production was suppressed by a change in the composition of the water (and likely its physicochemical properties, like surface tension and viscosity), which also changed the mechanics of diatom inclusion in the initial/sub-initial droplet.

2) The conclusion stated in that paragraph is again mechanistic. But likely the Reviewer might be right. However we are not in the position to provide more light on that particular point. On the other hand we have photos revealing cases of bubble rotary breaks that show even the cases of bubble overload with respect of collected cargo. In addition the cited work by Woolf and Trope (1991) and our research on bacteria (Marks et al., 2001) suggests that the conclusion stated is right. Considering the Reviewer suggestion the following is inserted:

3) [In addition, a build-up of microlayer at the water/air interface might suppress the efficiency of water jet formation by bursting bubbles.] 1) Line 240: It is difficult to conceptualize the relationship between bubble rise speed and the kinetic energy of the aerosol resulting from the bursting event. These are more likely separate processes, where the kinetic energy in the bubble rise is dissipated into the interfacial tension of the fluid at the air sea interface, and/or translational/rotational degrees of freedom of the bubble itself that are not related to the fluid fragmentation process of the bubble bursting event. Bubbles may reside at the water surface for a time before bursting, for lengths of time that depend on various parameters. The authors' explanation is difficult to verify based on the given information. In addition, the ability of a bubble to eject a cell is likely not limited by the energy required for ejection. The density of a cell is very close to that of the surrounding seawater (while it is acknowledged that cells are known to adjust their density to control buoyancy). The energy required to eject a diatom-free droplet is therefore likely very similar to the energy required to eject a diatom-laden droplet, although physical quantities to describe this difference are not known to this reviewer.

2) We observed that majority of bubbles did not rested (resided) at the interface. On the other hand sub-bubble vortex might also need a bit of time to rise-up and skim biological particles.

1) Suppression of jet droplet production has been shown in seawater enriched in diatoms and their associated dissolved organic matter when foams form on the water surface (e.g., King et al., Environ. Sci. Technol. 2013; Collins et al., Atmos Meas Tech 2014). It is important to consider the diatoms while including the dissolved organic carbon that they produce routinely. This comment also relates to the discussion on lines 328-338.

2) The comment is right, for conditions of foam formation at the water surface; however during our experiments foam was not formed. Therefore suggested references (e.g., King et al., Environ. Sci. Technol. 2013; Collins et al., Atmos Meas Tech 2014) are not included.

1) Figure 4: The conceptual framework of this figure is confusing. Consideration is not given to the hydrodynamic features of the diatoms themselves, yet such consideration is given to the much smaller ionic species. An explanation of diatoms existing at the base of a droplet may be associated with hydrodynamic drag during bubble ascent, where the larger particles migrate to the base of the bubble. More importantly, Phase C seems to suggest that the bubble bursting event is driven by the production of a cationic jet prior to film cap rupture. Such a concept is inconsistent with the known mechanism for bubble bursting. The detailed treatment by Lhuisseur and Villermaux (J. Fluid. Mech. 2012) [along with others prior and since] shows that the formation of the jet is related to the collapse of the bubble cavity following film cap rupture. This figure and associated conceptual model is therefore extremely confusing at best and may include fundamental flaws.

2) The reported research and concept dealt with observations conducted in saline water. The work by Lhuisseur and Villermaux, J. Fliid Mech. 2012 entitled "Bursting

bubble aerosol" refer to tap water bubbles and aerosol, therefore is not consider in interpretation of our work.

1) Table 1: If the initial/sub-initial droplets and secondary droplets are distinguished by ejection height, how was a lack of enrichment factor derived for secondary droplets. Any collections at low ejection heights would necessarily require the collection of initial/sub-initial droplets as well by consequence. Please clarify the method used to derive these quantities and provide data in plots and/or in entirety as supplemental information.

2) Experiments were designed to estimate the enrichment and to collect unbreakable diatom cells; therefore the estimation of enrichment in each droplet in jet set was not possible. However, after the main experiments we conducted research on diatom content in the top jet droplets. The results showed that diatoms behaviour was different as compared with bacteria. The rule reported by (Blanchard and Syzdek, 1978) that "the concentration of bacteria is always highest in the top jet drop of the jet set and decreasing progressively in the lower drops" was not confirmed for diatoms. Therefore we decided to refer rather to enriched initial/sub-initial jet droplets. Following that suggestion a rephrased paragraph and a new Fig. 5 (see below) is inserted.

3) [Blanchard and Syzdek (1978) determined that the concentration of bacteria is always highest in the initial jet droplet and decreases progressively in the lower drops, being lowest in the last ejected droplet. That however was not exactly the case with ejection of diatoms. Thus, at this stage of research we may only state that a single bubble at the moment of burst projects diatoms which are incorporated to one-two initial/sub-initial jet droplets of five-seven jet droplets produced by each jet.]

1) Table 2 shows that the range of the number of diatoms existing in enriched droplets widens by a factor of 2 for the more concentrated sample. A listing of the statistical parameters of the distribution of observations from these experiments would substantially improve the clarity of these results in the context of the discussion within the text.

Providing a histogram of observations (e.g., a frequency distribution of observing 'N diatoms per droplet') would help mightily. 2) The experimental evidences showed rather high variability and a tendency of decreasing (drifting) of enrichment values in elapsing time of aeration thus we decided to express our data as averaged, without more expanded statistics. We also learned that each bubble is different, especially if rising in water containing suspended matter and biological particles. Considering the Reviewer comment a new Figure 5 that show number distribution of droplets with diatom cells, without diatoms and number distribution of diatom cells in each sample is inserted into altered manuscript. The following text and a new Fig. 5 were inserted:

3) [An example of typical data distribution and related variability is show in Fig. 5, where number of droplets with and without diatom cells as well as number of diatom cells in each of 20 samples is plotted. Data refer to experiment with suspension of Nanofrustulum cells aerosolized by uniform bubbles of D = 1.2 mm rising in water column containing 840000 cells/mL under 22.1oC temperature and 35 g/kg salinity.] [Figure 5] 1) Line 357-363: This conceptual framework (paired with Figure 4) may be inconsistent with the current paradigm based on detailed observations by multiple investigators – or the explanation may be unclear. A greater defense of the proposed mechanism should be provided and/or the authors' argument should be presented in light of the current evidence-based understanding of the bubble bursting mechanism in which the basal jet is formed during bubble cavity collapse, and not associated with the bubble stopping as it approaches the (bulk) air-water interface.

2) I am sorry to say that for bubbles D = 1.2 mm there is nothing like the bubble cavity collapses alone and related linear projection of jet droplets. The jet droplets are projected upward mainly by vortex, which is developing below bubble bottom half sphere. Recently we collected even photos of single jet droplet d = 0.12 mm that is spiralling upward. Other experiments indicated that jet droplets collected on "rotary detection disc" made of porous and light foam develop counter-clockwise rotational motion in the Northern Hemisphere. That again points to dominating role of bubble rotational

features and particular significance of sub-bubble vortex.

1) Line 365: In order to support this argument, please cite evidence to suggest that diatom cells are more dense than seawater and bacteria cells. Line 367: Diatoms are indeed larger than bacteria, and therefore are more subjected to hydrodynamic drag as bubbles ascend through the water column. This simple argument (as discussed above) would provide an equally valid case for inclusion of diatoms in the initial/sub-initial droplets that are derived from the jet. It is important not to forget that the concentration of bacteria in the ocean also dwarfs the number concentration of diatoms by multiple orders of magnitude, so the statistical likelihood of finding a bacterium in a droplet is simply greater.

2) The statement base on observations and cited above research reported by (Blanchard and Syzdek, 1978). However, to assist Readers (Discussion, 2 paragraph) is altered:

3) [The airborne fate of diatoms might depend mainly on cells weight as well as wind speed and air humidity conditions. Our laboratory observations indicate that diatom cells may even break into fragments during drying, which could be then easily dispersed in the troposphere under high or moderate wind speed conditions (e.g., Bigg and Leck, 2005).] 1) Line 403-410: This discussion is off topic and confusing. Define a 'spiraling chain of cations'. What sort of cations are involved? This is particularly hard to envision with inorganic ions. How is this related to the topic of the manuscript? Where are the DNA and RNA bases coming from? This paragraph could be easily omitted without a loss of context or substantive discussion of the results. In general, the discussion of the cationic vortex and related items are largely irrelevant to the stated motivation of the study. The discussion of this topic is also confusing and not well buttressed by evidence. It is interesting to think about the charge separation that reportedly occurs in the rising bubble, but the deeper mechanistic discussion strays from the important point.

2) The Reviewer statement that "the cationic vortex and related items are largely irrelevant to the stated motivation of the study" is not accepted. Explanation is phrased above (lines 357-363).

1) Line 446-453: Much greater detail on the methods and findings (including visualized data!) that reflect the description in this paragraph should be included in the manuscript.

2) More expanded explanation is given above and refer to (line 198). Visualization of data distribution is plotted in Fig. 5, see responses above.

1) Line 507: The conclusions relating to differences in concentration should be tempered, as the mechanism may not be well explained or explored by this study, and only two concentrations were attempted.

2) As suggested by the Reviewer the conclusion was tempered to read:

3) [The enrichment factors for aerosolized diatoms range from 8 to 307 and were found to decreases with increasing concentration of suspended diatoms in the water column.] 1) Line 513: The claim in the last sentence is almost entirely unrelated to the study at hand, and at the same time, is a significant stretch in logic. Prebiotic chemistry is, indeed, an exciting field, but the simple fact that a vortex exists beneath a rising bubble does not necessarily suggest that this is related to proto-nucleic acid formation – although in making this argument, it is clear to see that this subject of the detailed structure of cationic vortexes is quite off topic for the current manuscript.

2) I strongly feel that this is the case, especially considering a double helix like spiraling motion developing by rising bubbles. For the first time we are in the position to explain the (mechanistic) rotational/ionic mystery of proto-nucleic formation. The chemistry and microbiology parts of that puzzle may fit and very likely can be now placed onto the double-helix bi-ionic motion of bubbles. Therefore I would like to keep that statement as is.

Other comments 1) Line 132: "bubble mediated scavenge [sic] and aerosolization of biomolecules with special attention given to diatoms" The authors perhaps mean to say 'aerosolization of biomaterials' as diatoms are not cells and not molecules. The proper terminology for diatom morphology should be used: Cyclotella are 'centric' diatoms, while Nanofrustulum are 'pennate' diatoms.

2) As suggested "elongated" and "spherical" were replaced by "pennate" and "centric" in the text and in Fig. 1.

3) [Figure 1. Stains of diatoms used in experiments: A) pennate Nanofrustulum and B) centric Cyclotella. Picture taken by Canon DS 500D using Zeiss Scope A1 with PlanApo x100 lens.]

1) Diatoms have been referred to in this manuscript in general terms using "bio-cells" or "bio-matter". These are not accepted terms and the authors should simply refer to them as "cells", "biological material", or even "biological particles", as are often used in the oceanography literature when referring to filter-retained material in the ocean. Concentrations should be provided using standard unit notation (cells/mL) rather than noted as "cells / 1 mL." In tables, units of volume (mL) are consistently given instead of units of concentration. It is suggested that the authors seek further editing of English usage and grammar for this manuscript.

2) Thank you for that comment, accordingly the whole text was reviewed and expressions like "bio-matter" or bio-molecules" were replaced by [biological particles] or simply [cells]. Also the suggested corrections regarding concentration units (cells/mL) is appreciated. In addition, further English corrections were conducted. 3) Units 'cells/mL; mL' and expressions "biological particles"; "cells" were inserted in a new version of the manuscript.
* * *
[Figure]

**Figure 5.** Number of droplets containing *Nanofrustulum* diatom cells, without cells and number of cells in each sample aerosolized by bubbles of $D$ = 1.2 mm.

**Fig. 1.**

---

## Author Comment (AC2) · 18 Jan 2018

Refer to: Ocean Science: os-2017-82, submitted on 03 Oct 2017 Manuscript Number: PAAG-D-14-00412R2; Title: Rising bubbles as mechanism for scavenging and aerosolization of diatoms; Authors: Roman Marks, Ewa Górecka, Kevin Mc Cartney, and Wojciech Borkowski Responses to the Anonymous Referee #2

First of all I would like to thank the Reviewer for many valuable comments. Base on them a new version of the manuscript was prepared. 1. Comment.The manuscript presents some interesting postulations on the mechanisms for the scavenging of diatomaceous material by rising bubbles and the aerosolization of scavenged material on bubble-bursting at the sea-air interface. I am reviewing this from the perspective of

marine atmospheric science, where there is a substantial body of literature concerned with the bubble-mediated production of sea-spray aerosol. Convergence of interpretations of evidence in this field will be extremely important in resolving outstanding mechanistic uncertainties and the current work aims to provide mechanistic insight into this debate. However, as it is written, I have some general problems in understanding the concepts and implications of this work and there are quite a few points that must be addressed before the manuscript is publishable.

2. Response. The literature concerned with bubble-mediated sea spray production is expanded in altered manuscript and include more recent references from marine atmospheric sciences. Our research on the bubble scavenge and aerosolization is based on experimental evidences that revealed ability of rising bubbles to develop bi-rotational and bi-ionic-electrical features, which are mechanistic and may be directly used to explain the principles of bubble mediated scavenge of bio-particles in the water column and aerosolization. The main evidences were gained by tracing photography records of rising bubble. The experiments require proper positioning of light source below the cylinder or at top, and more importantly the water has to be relatively clean (filtered) as well as relatively warm. The rotary tracers become fogy and not distinguishable when water contains too much of suspended materials. Since two distinct rotational systems were identified, we realized that also a charge polarity within the bubble upper and bottom half spheres is gathered during the bubble rise. The results are presented in (Marks, 2014; 2015). This is why we explain the diatoms/bacteria aerosolization base on both: the mechanistic and electrostriction principles. Considering the Reviewer comments, a new version of manuscript has been prepared and content of Literature expanded, see responses below.

1. Context: First, I have a couple of points about the context of the work. Much atmospheric work has focussed on the enrichment of organic material in seaspray particles. Owing to the much greater number of particles in sizes from around one to a few hundred nm and the importance of these as potential cloud nuclei, much of the work has

[Figure]

concerned characterisation of organic material in these particles in terms of its broad chemical functionality and influence on water uptake in the moist atmosphere (O'Dowd et al., 2004; Wang et al., 2017). As recognised at the top of section 1.3 in the current manuscript, primary marine particles in these size ranges are from the "film" mode, formed when the bubble cap shatters on bursting, not from the larger sized, but less abundant, "jet" mode of particles. It has been shown that enrichment of diatomaceous exudate in this film mode is of more atmospheric significance than that of organic material of other phytoplankton origin (Fuentes et al. 2010), but there has been less recent atmospheric focus on aerosolised intact diatoms since they will generally be too large to be found in the abundant film particle mode and will not thereby influence liquid cloud droplets. However, there has been recent interest in the potential roles of diatoms as the much scarcer ice nucleating particles (Wilson et al., 2015). For this reason (and for mass budgeting and biogeochemical cycling purposes, as recognised in the few recent studies in this area), mechanisms for their atmospheric aerosolisation will be important (particularly under low wind speed conditions relevant for the Arctic atmosphere). I believe these contextual considerations are important to set the scene for the current work. However, once the atmospheric context is appreciated, the relevance of enrichment of diatoms in droplets of 120 microns diameter becomes apparent and the likelihood of such enrichments playing a role on atmospherically important timescales must be seen as negligible.

2. The organic material in sea spray particles are aerosolized by both film and jet modes. The airborne dispersion of film mode is instant when bubble bursts, but particles accommodated to less abundant jet mode (although dominating in term of mass) contributes, since jet droplets may dry and undergo fragmentation. For example diatoms drying in droplets may even "explode" and break to fragments. These fragments are likely more dispersible and influence condensation of water vapour and "water uptake in the moist atmosphere". In addition list of literature is expanded.

3. [The importance of aerosolized organic materials as cloud condensation nuclei and

their broader atmospheric functionality is reviewed by (O'Dowd et al., 2014). Apparently, also the contribution of diatomaceous is of more atmospheric significance than the other organic materials of a phytoplankton origin (Fuetnes et al., 2010). It has been also suggested that fragments of diatoms may provide ice nucleating particles (Wilson et al., 2015).]

[Bigg C., Leck E.K. (2005) Biogenic particles in the surface microlayer and overlaying atmosphere in the central Arctic Ocean during summer, Tellus B, 57,4, 305-316, DOI: 10.1111/j.1600-0889.2005.00148.x.

Lee C., Sultana C. M., Collins D. B., Santander M. V., Axson J. L., Malfatti F., Cornwell G. C., Grandquist J. R., Deane G. B., Stokes M. D., Azam F., Grassian V. H., Prather K. A. (2015) Advancing Model Systems for Fundamental Laboratory Studies of Sea Spray Aerosol Using the Microbial Loop. J. Phys. Chem. A, 119338860–7010.1021/acs.jpca.5b03488.

Lakhina G.S. (1993) Electrodynamic coupling between different regions of the atmosphere. Current Science, 64, 660-666.

O'Dowd C. D., Facchini M. C., Cavalli F., Ceburnis D., Mircea M., Decesari S., Fuzzi S., Yoon Y. J., and Putaud J.-P. (2004) Bio- genically driven organic contribution to marine aerosol, Nature, 431(7009), 676–680.

Patterson J. P., Collins D. B., Michaud J. M., Axson J. L., Sultana C. M., Moser T., . . . Gianneschi N.C. (2016) Sea Spray Aerosol Structure and Composition Using Cryogenic Transmission Electron Microscopy. ACS Central Science, 2(1), 40–47. http://doi.org/10.1021/acscentsci.5b00344.

Wang X., Deane G.B, Moore K.A., Ryder O.S., Stokes M.D, Beall C.M., Collins D.B, Santander M.V., Burrows S.M., Sultana C.M. and Prather K.A. (2017) The role of jet and film drops in controlling the mixing state of submicron sea spray aerosol particles, PNAS 114 (27), 6978-6983, doi:10.1073/pnas.1702420114.

Wilson T.W., Ladino L.A., Alpert P.A., Breckels M.N.B., Brooks I.M., Browse J., Burrows S.M., Carslaw K.S., Huffman J.A., Judd C., Kilthau W.P., Mason R.H., McFiggans G., Miller L.A., Nájera J.J., Polishchuk E., Rae S., Schiller C. L., Si M., Temprado J.V., Whale T.F., Wong J.P.S., Wurl O., Jakobi-Hancock J.D.Y., Abbatt J.P.D., Aller J.Y., Bertram A.K., Knopf D.A.K., and Murray B.J. (2015) A marine biogenic source of atmospheric ice-nucleating particles, Nature, 525, 234–238, https://doi.org/10.1038/nature14986.]

1. Abstract: It is not clear to me that the abstract serves the usual purpose of an abstract. Almost all the text is background information that does not summarise the findings of the paper and refers to previous literature (mainly from the first author). Indeed, the only part of the abstract that relates specifically to the paper findings is the last sentence. 2. Accepting the Reviewer comment (Abstract) is shortened. Two first sentences are deleted and third altered to read:

3. [Abstract. Bubbles rising in relatively clean saline water cause steady displacement of ions at the bubble boundaries which creates electric polarity that draw biological particles dispersed in the water. Viable diatoms as well as bacteria develop negative charge on outer membranes that are attracted to the cationic bubble bottom half sphere. When bubble bursts at the air/water interface the diatoms and bacteria are ejected into the air with initial or secondary jet droplets that are projected upward with a small water column derived from a sub-bubble cationic vortex. Laboratory experiments conducted in brackish and oceanic saline water on Nanofrustulum and Cyclotella cells indicate that the average concentration of jet droplets compared to original water volume (here termed the enrichment factor) for aerosolized diatoms may range from 8 to 307.]

1. Introduction: I found this rather meandering and not all directly relevant to the current work. The last paragraph of section 1.3 outlines the findings of previous experiments on the dynamics of bacterial scavenging by rising bubbles and states that the novelty of the current study is "...the bubble mediated scavenge [sic] and aerosolization of

bio-molecules with special attention given to diatoms".

2. Introduction is shortened and reedited. In addition a new paragraph underlying contribution of atmospheric researchers to study the sea spry aerosol is inserted. Also the Reviewer comment related to section 1.3 is rephrased:

3. [Increasing interest in biological and chemical nature of sea salt aerosols has resulted in numerous field experiments and papers (e.g., Bigg and Leck 2005; Patterson et al., 2016). In particular the role of sea spray particles in determining electrical mobility of particles and related cloud properties was investigated by (Wang et al., 2017). In addition, evidences that aerosol samples contains phytoplankton and fragments of diatoms were reported (e.g., Bigg and Leck, 2005; Lee et al., 2015; Patterson et al., 2016). Furthermore, the importance of organic materials as cloud condensation nuclei and broader atmospheric functionality is reviewed by (O'Dowd et al., 2014). Apparently, the contribution of diatoms is of more atmospheric significance than other organic materials of a phytoplankton origin (Fuetnes et al., 2010). It has been also suggested that diatoms fragments may provide ice nucleating particles (Wilson et al., 2015). However, no particular investigations on enrichment of diatom cells in jet droplets have been reported.]

[The present research further explores the bubble mediated scavenge and aerosolization of biological particles with special attention given to diatoms.] 1. Methodology: I have a number of queries about the experimental procedure. i) what is the air pump flow rate? This would be useful to understand the scavenging dynamics. 1 sccm would be 1.667x10Ȩ̈-5 L/s. Each 1.2 mm diameter bubble is 0.905x10Ȩ̈-6 L, so this would be 18.42 bubbles per second (@ 1 sccm), so 3 seconds of bubbles with 7 jet drops per bubble would be 387 droplets collected at this flow rate. At a droplet diameter of 0.12 mm, this is 3.5 x 10Ȩ̈-7 L = 0.35 microlitre. ii) how was the bubble size measured (optically or acoustically)? iii) how was the droplet size measured? iv) was the RH actively controlled to ensure the water evaporation rate was constant for all experiments and hence the diatom concentration was being measured at the same water activity

(and hence normalised to ionic strength) Given the 150 ml volume, I do not understand how sampling roughly of the order of 1/3 microlitre in 3 seconds can substantially lead to "both diatoms concentrations in the water suspension and water salinity (or more precisely the availability of cations) decrease during aeration". Obviously the air pump flow rate could be very much higher, but then this would lead to many thousands of droplets on the slides.

2. The air flow rate was $\sim$ 0.35 mL/s; however the main attention was given to produce separated bubbles that were not forming foam and to collect separated droplets under highest possible air humidity, needed for diatoms counting. That was achieved by covering the glass breaker by microscopy slides during the aeration. Then slides were immediately closed in Petri Dish with small volume of water, in order to avoid the damage of diatoms. In closed Petri Dish air humidity was 100%. The sizes of bubbles were measured optically by sizing the photo-tracers of bubbles in the water or at the surface as well as tracers of jet droplets just after ejection into air. In order to collect set of samples during each run, much longer time was needed then 2-3 seconds. Considering the reviewer comment the expression (cation availability) was deleted and accordingly the first sentence in page 7 was altered. In order to provide more information on electric charge distribution around bubbles we inserted a following:

3. [In order to detect electric charge distribution around the rising bubble boundaries an oscilloscope, type HANTEK$^{®}$-DS01201BV was used, allowing to trace the voltage polarity between a two probe heads in mV with 0.1 mV detection limit and 0.5% accuracy. One head was submerged to the depth of 2 cm in the bulk water, and the second probe tip was placed just at the water/air/bubble interface. The measurements were conducted in a small glass tank filed with 700 mL of saline (35 g/kg) water of 21.0oC temperature. Bubbles D = 1.2 mm were generated by capillary aerator with a rate allowing ca. 20 mm distance between a successive bubbles. These rose about 6 cm in a free water column and then were slowed on a glass plate tilted to 30o that reduced a zig-zag motion of bubbles, allowing interactions with the tip of probe head.

The measurements showed negative charge introduced by bubbles outer layer that typically adhered to the tip at water/air/bubble interface. However, from time to time also strong positive charge was measured, indicating interaction with cationic domain gathered within surfacing bubble vortex. The maximum negative and positive values were considered as revealing polarity of ions gathered around the bubble upper and bottom half spheres.] 1. Consequently, I am not sure of the statement on lines 194-196: "after analyzing that set of data we noticed that the efficiency of diatom aerosolization increased, indicating that rising bubbles reduced concentrations of diatoms (solution was gradually cleaned) during the experiments."

2. Thank you for that comment the word (increased) was replaced by (decreased). The sentence was rewritten:

3. [However, after analyzing that set of data we noticed that the efficiency of diatom aerosolization decreased in time of aeration, indicating that rising bubbles reduced the concentrations of diatoms in the water penetrated by bubbles (solution was gradually cleaned). In addition, the development of microlayer and changes in physicochemical properties of the water/air interface gradually suppressed the bubble mediated diatoms aerosolization.] 1. Results: My most significant concern is with the presentation of the results - or more precisely, the lack of systematic presentation of results. The results should be detailed along with errors and statistical distributions to support the hypotheses. The descriptions of the results is hard to follow as it is currently written, since it is frequently mixed with discussion material (e.g. lines 219-222, 230-232, 240-247). When this discussion is removed, it become apparent that there are few results presented.

2. Following the Reviewer comment, the first paragraph and in particular (lines 219-222) are rephrased. Lines 230-232 are corrected. In addition, the context of Lines 240-247 with expanded explanation is inserted to that paragraph.

[Considering that a single burst of bubble D = 1.2 mm may eject about 4-8 droplets from

a clean water at temperature (Tw) of 20oC (Blanchard and Syzdek, 1982), the number of jet droplets that contained diatoms were counted and compared with droplets that lacked diatom cargo. The comparison showed that only 20-25% of jet droplets were enriched by diatoms, which suggests that the diatom aerosolization process might be influenced by a combination of factors operative in the water column, or at air-water interface. To determinate whether the diatoms concentration in the top jet droplet is always highest, as reported for bacteria (Blanchard and Syzdek, 1978), jet droplets were collected at different elevations above the water level with special attention given to initial droplet. The observations indicated that not exclusively top jet droplets were diatom enriched.] [Since diatom cells are relatively large, as compared with bacteria cells, these are more resistant to sudden displacement with cationic jet, thus contribute into sub-initial droplets.]

[In addition, a build-up of microlayer at the water/air interface might suppress the efficiency of water jet formation by bursting bubbles.] 1. Table 1 refers to very few droplets, from which I cannot understand the methodology (number of repeats, flow rates and sampling strategy etc...) Moreover, many of the the discussion points are either incorrect of logical non-sequiturs (e.g. line 239 - there is no reason for an inability of bubbles to scavenge diatoms to lead to a reduction in jet droplets and line 258 - there is no reason that an enrichment factor indicates that diatoms were from the bubble bottom layer, though they may have been). 2. Accepting the Reviewer comment the number of samples collected during each set of experiments was inserted to Table 1 and 2. 3. (Table 1) [(N) number of samples 20 8 ] (Table 2) [(N) number of samples 20 20 ]

1. I am not sure that I follow the attempt to distinguish the initial and secondary droplets through the ejection heights as outlined in lines 253-257 when it is clearly stated that there was "not exclusively top jet droplets were enriched by diatoms" on 219 and "Conducted screening showed that only 20-25% of jet droplets were enriched by diatoms, which suggests that the process of diatom aerosolization might be influenced by a

combination of factors operative in the water column, or at air-water interface". I would have liked to have seen some statistical analysis of the distribution of enrichments with height (for example) to support the mechanistic contentions and postulations.

2. In order to investigate the distribution of diatoms with droplets high, the other set up (probably modification of that used by Blanchard and Syzdek, 1978, would be required). We designed our experiments to estimate the enrichment factor for aerosolized diatom cells of two shapes and considering two suspensions. However, I do agree that much more research is needed and in particular exploration by methods that are in hands of chemist might provide new evidences. I strongly feel that these will support our findings. Additional explanation and graphical description of droplets and diatoms distribution was included according to the Reviewer postulates.

3. [Figure 5] [An example of typical data distribution and related variability is show in Fig. 5, where number of droplets with and without diatom cells as well as number of diatom cells in each of 20 samples is plotted. Data refer to experiment with suspension of Nanofrustulum cells aerosolized by uniform bubbles of D = 1.2 mm rising in water column containing 840000 cells/mL under 22.1oC temperature and 35 g/kg salinity.] 1. Discussion & Conclusions: The discussions section presents some interesting conjecture, but does little to draw on and add to the results of the paper. Furthermore, I do not believe that the authors have presented results that can be evaluated sufficiently rigorously to support the stated conclusions.

2. Both sections include more information and were reedited.

1. Minor The sentence starting on line 48 reads peculiarly. The statement "generated at wind velocity 8 m/s" makes the sentence appear to be a non-sequitur, since the previous sentence states simply that the lume depth is the same as the wave height (with no wind speed dependence). If this is simply additional information, the sentence can straightforwardly be rephrased. The second phrase needs further punctuation "while, during rain.

2. Thank you for that remarks. The sentence on line 48 was shortened and punctuation (while, during rain) inserted to read:

3. [A typical breaking wave generates a downward circulating rotor–like motion reaching a depth equal to the wave height (Thorpe, 2001) while, during rain bubbles may occur to about 10-20 cm depth (Blanchard, 1963; Katsaros and Buettner, 1969).] 1. Line 66 "Cl/Na atomic mass..." add "RATIO" 2. The "ratio" was added. 3. [Using a Cl/Na ratio of atomic mass equal to 1.542 (Kropman and Bakker, 2001)...]
* * *
[Figure]

**Distribution of droplets and diatom cells**

**Figure 5.** Number of droplets containing *Nanofrustulum* diatom cells, without cells and number of cells in each sample aerosolized by bubbles of $D = 1.2$ mm.

**Fig. 1.**